# The Clinical Feasibility and Safety of 1.5 T MR-Guided Daily Adapted Radiotherapy in 1000 Patients: A Real-World Large Experience of an Early-Adopter Center

**DOI:** 10.3390/cancers17122012

**Published:** 2025-06-17

**Authors:** Chiara De-Colle, Michele Rigo, Andrea Gaetano Allegra, Luca Nicosia, Niccolò Giaj-Levra, Edoardo Pastorello, Francesco Ricchetti, Carolina Orsatti, Andrea Romei, Nicola Bianchi, Riccardo Filippo Borgese, Antonio De Simone, Davide Gurrera, Stefania Naccarato, Gianluisa Sicignano, Ruggero Ruggieri, Filippo Alongi

**Affiliations:** 1Department of Advanced Radiation Oncology, Cancer Care Center, IRCCS Sacro Cuore Don Calabria Hospital, 37024 Negrar di Valpolicella, VR, Italy; chiara-de-colle@sacrocuore.it (C.D.-C.); michele.rigo@sacrocuore.it (M.R.); andrea.allegra@sacrocuore.it (A.G.A.); niccolo.giajlevra@sacrocuore.it (N.G.-L.); edoardo.pastorello@sacrocuore.it (E.P.); francesco.ricchetti@sacrocuore.it (F.R.); carolina.orsatti@sacrocuore.it (C.O.); andrea.romei@sacrocuore.it (A.R.); nicla.bianchi@sacrocuore.it (N.B.); riccardo.borgese@sacrocuore.it (R.F.B.); antonio.desimone@sacrocuore.it (A.D.S.); davide.gurrera@sacrocuore.it (D.G.); stefania.naccarato@sacrocuore.it (S.N.); gianluisa.sicignano@sacrocuore.it (G.S.); ruggero.ruggieri@sacrocuore.it (R.R.); filippo.alongi@sacrocuore.it (F.A.); 2University of Brescia, 25121 Brescia, BR, Italy

**Keywords:** MR-Linac, adaptive radiotherapy, MRI-guided radiotherapy, online plan adaptation

## Abstract

MR-guided radiotherapy using MR-linacs (MRL) has seen rapid growth in recent years. Daily online plan adaptation and real-time motion monitoring offer key advantages across various tumor sites. However, concerns persist regarding extended treatment durations and restricted patient eligibility, highlighting the need for continued optimization and careful patient selection in clinical practice. This study reports the clinical experience of a 1.5T MRL early-adopter center. Data from 1000 patients were analyzed to assess feasibility, treatment duration, and toxicity. With a majority of prostate treatments and other common sites including lymph nodes, pancreas, and liver, three-quarters of the treatment sessions could be completed within 40 min (mean in-room time: 38 min). Grade 3 toxicity rate were very low, with no higher-grade events. These findings support the feasibility and safety of MR-guided radiotherapy across multiple tumor sites, with ongoing efforts to optimize protocols for dose escalation and motion management strategies.

## 1. Introduction

Magnetic resonance-guided radiotherapy (MRgRT) on MR-linacs (MRL) represents one of the major technological advances over the past few decades in radiation oncology. The superior soft-tissue contrast compared to computer tomography, together with the possibilities of daily MR-guided plan adaptation and organ motion management strategies for real-time MR-guided target gating, allow for extreme accurate and personalized treatments [1,2,3]. These advantages have been exploited for radiooncological treatments in different tumor sites, such as prostate, pelvic lymphnodes, rectum, liver and others [4,5,6,7,8,9]. Superiority of adaptive strategies compared to conventional (non-adaptive) radiotherapy workflow in terms of target coverage and organs at risk (OARs) avoidance has been documented for pelvic and abdominal tumors [10,11,12,13], with preeminent evidence for MRgRT applied to prostate cancer [14].

However, some concerns regarding treatments at the MRL, which might hamper its clinical applicability and minimize the impact of its benefits, still remains. These include the increased treatment time due to the MR acquisition and online adaptive workflow and eventually the limited patient eligibility due to clinical or technical aspects, e.g., the smaller treatment field size [12,15].

In the present manuscript we present our experience with MRgRT at the 1.5T MR-linac Elekta Unity (Elekta AB, Stockholm, Sweden) on a large cohort of 1000 patients treated in our department from the moment of the MRL installation and its clinical application in 2019. We extensively report on clinical and technical aspects, such as indications, adaptive strategies adopted, treatment times and toxicity.

## 2. Material and Methods

The first 1000 consecutive patients treated at the MRL in our department were arbitrarily selected for the present analysis. Between October 2019 and May 2022, patients were recruited into a prospective observational trial (Prot. N° 2155CESC). Since March 2023, patients have been treated within the MOMENTUM protocol (Prot. N° 4164CESC). Between June 2022 and February 2023, patients’ data have been prospectively collected outside a clinical protocol, upon patients’ signature of the informed consent for data analysis for research purpose.

### Treatment Characteristic

Treatment workflows for MRgRT at the 1.5T MRL have been previously described [4,16]. Briefly, patients underwent both simulation MR with 1 mm slice thickness and simulation CT (Somatom AS, Siemens, Germany) with 3 mm slice thickness, the latter for dose-calculation purpose. For prostate/prostate bed treatments, patients were instructed to have half-full bladder and empty rectum. A T2-weighted MR scan was acquired for simulation and prior to each fraction. For treatments in the upper abdomen, patients were instructed to fast 3 h before RT and drink half a glass of water right before MR simulation and the treatment session, in order to better visualize the stomach and duodenum in the T2 sequences. Here, additionally to the latter, T1-weighted sequences as well as navigated and 3D Vane sequences were acquired during simulation. The fractionation choice was mainly made according to RT site, oncological situation and target/organs at risk anatomy, e.g., for postoperative prostate treatments eventually including pelvic irradiation as well, slightly hypofractionated schedules were adopted, for prostate cancer with low/intermediate risk and low IPSS and a prostatic volume less than 80 cc, a 5-fraction schedule was preferred. For upper abdominal targets, i.e., pancreas and liver, 45–50 Gy in 5 or 6 fractions were applied, unless because of target’s size, number of treated lesions and/or anatomical reasons less hypofractionated schedules were needed. Margins used for the PTV were 5 mm isotropic, except for 3 mm posteriorly for prostate/prostate bed treatments. The software Monaco (Elekta AB, Stockholm, Sweden) was used for treatment planning. The dose was prescribed according to ICRU guidelines [17]. Dose constraints for prostate stereotactic treatments were previously described [4,18], while for stereotactic RT in other sites Timmerman’s tables were considered [19].

Technical and treatment information such as radiotherapy schedule, treatment time and adaptive technic have been prospectively recorded, while toxicity data were retrospectively collected. Regarding the technical and treatment information, within the radiotherapy schedule we recorded dose and fractionation as well as delivery in consecutive days or every other day. The treatment time represents the overall treatment time, including patient positioning, MR acquisition, image fusion, recontouring when adapt-to-shape (ATS) method was used, replanning and dose delivery. We recorded these data prospectively for every fraction and report here the average treatment time per patient. Specifically, for dose delivery, in September 2023 in our department the comprehensive motion monitoring (CMM) was installed. Compared to the until then used motion monitoring (MM), this tool allows not only for real-time target visualization, but also for gating and eventually replanning on the basis of a new isocenter to correct for an intrafractional drift [20]. CMM MR images are available as T1, T2, balanced, FLAIR and SPAIR, whereas the most used in our department are the new T2-weighted Turo Spin Echo sequences for prostate and pelvic located targets and balanced for upper abdomen targets. In order to identify the target on cine images, two template images are firstly generated, with an algorithm very robust for tissue contrast, and registered to the daily 3D MRI. After revision and template approval, cine real time images are acquired using a fast protocol and registered against the template images, having these the same contrast. The target position is real-time compared to the adapted plan. The following CMM structures are identified: (1) the registration structure, which is used by the registration algorithm, and (2) the gating envelope, which encompasses the target, with a margin expansion. A tolerance level is defined and, whenever the gating envelope structure moves outside the registration structures above the tolerance, the beam is disabled and irradiation stops, restarting automatically when the target is again within the tolerance level position. Usually, we defined the GTV or the CTV (e.g., the prostate gland) as registration structure and the PTV as gating envelope. During dose delivery, according to different scenarios, multiple actions can be undertaken, namely: (1) to lower the tolerance lever, (2) to generate a so-called baseline-shift plan or (3) to disable the CMM. Specifically, lowering the tolerance level allows for an increase in the beam hold time, at the cost of lowering treatment precision, while, when a stable shift in the target position occurs, a baseline-shift plan is calculated on the basis of the new position. This is a quickly generated plan (within 1–2 min), which takes into account the target coverage, but not all the constraints for the OARs that were implemented in the original or adapted treatment plan. The remaining monitor units are then delivered with this new plan. Finally, disabling completely the CMM leads to the classic motion monitoring strategy, where live MR images with the target structure are visible, but no gating occurs.

As for the adaptive technic, we describe for each fraction whether an adapt-to-position (ATP) or ATS method [1] was adopted.

Median follow up was calculated from the start of radiotherapy. Acute and late toxicity were defined using the cut off of 90 days and according to the CTCAE scale Version 5.0.

## 3. Results

### 3.1. Patients Cohort

Between October 2019 and June 2024, 1000 patients were included for a total of 1061 treatment courses. Median follow up was 39 months (range, 0–60). Mean age was 69 years (range, 16–90). All treatments characteristics are summarized in Table 1. We irradiated the primary tumor or tumor bed in 72.6% of the cases, lymph node metastases in 16%, distant metastases in 7%, while 4.4% of the cases were retreatments. Prostate and prostate bed were irradiated in 57.1% and 10.2% of the cases, respectively, including pelvic lymphnodes in 4.7%. Other targets were represented by bone metastases and tumors of the pancreas, liver, brain, adrenal gland and lung. Less frequently represented targets were bladder, soft tissue sarcomas, vas deferens, anal/rectal cancers, head and neck, kidney and gynecological cancers. The most frequently prescribed doses were 36.25 Gy (31%), 35 Gy (28.3%) and 30 Gy (9.4%) in five fractions. Less frequently, moderate or normofractionated treatments were performed, i.e., 60 Gy in 20 fractions and 67.5 Gy or 66 Gy in 30 fractions, respectively, adopted in 7.5%, 5.6% and 1.7% of the cases. The two most commonly adopted treatment schedules according to RT site are listed in Appendix A. The large majority of the treatments (70%) were delivered in consecutive days, while a minority was delivered every second day (29%) or with other schedules.

### 3.2. Treatment Time

On a total of 9076 administered fractions, 80.8% were with the adapt-to-shape method and 19.2% were with the adapt-to-position method. The mean in-room time was 38 min (range, 18–103), with 74.4% of patients completing the session within 40 min. In 95% of the cases (1007 treatment courses), the mean in-room time was less than 60 min. Considering the 5% of the cases where the treatment lasted longer than 1 h as outliers, as they might represent real but extreme cases (Appendix A) and excluding them from the treatment time analysis, we observe that the mean in-room time 36 min. Considering the total of 1007 treatment courses, the average treatment time was 41 min in the first year and 33 min the last year (Table 2). Treatment times were longer for targets located in the upper abdomen, i.e., pancreas and liver, with mean treatment times of 41 and 39 min, respectively. Prostate targets, including prostate and prostate bed with or without pelvic irradiation displayed an average treatment time of 36 min. Targets which did not show high inter- or intrafraction variability, i.e., isolated lymph node metastases, bone metastases and cerebral tumors, could be irradiated, on average, in 33, 32, and 28 min, respectively (Table 3).

When considering the total of 1061 treatment courses, the mean treatment time was 44 min and 34 min in the last year (Appendix A) with longer treatment times for liver and pancreatic targets, namely 44 and 42 min, respectively, and shorter treatment times for lymph node metastases, bone metastases and cerebral tumors, namely 33, 32, and 28 min, respectively. Details of the treatment times, according to the treated sites, for the total of 1061 courses are listed in Appendix A. Because of the wide range of age, we conducted a sub analysis among the 66 treatments performed on patients older than 80 years. Here, the mean treatment time was 37.2 min.

### 3.3. Treatment Toxicity

Regarding acute toxicity, 61.9% patients developed no toxicity, 30.3% developed Grade (G) 1, 6.1% G2 and 1.6% G3 toxicity (Figure 1A), with no patients experiencing toxicity >G3. On a total of 858 patients available for late toxicity analysis, no toxicity was recorded in 91% of the patients, G1 in 6.8%, G2 in 6.1% and G3 in 0.3%, with no >G3 (Figure 1B). As we predominantly performed treatments of pelvic- and abdominally located tumors, genitourinary and gastrointestinal were the most frequently recorded symptoms (Table 4 and Table 5). Among other toxicities we recorded asthenia, abdominal pain, mucositis and erythema. Analyzing the incidence of G2/G3 toxicity among the patients that were treated in less than 38 min (our mean in-room time) compared to the patients that were treated in 38 min or longer, no difference was found (7% vs. 9%, *p* = 0.1). As we did for the treatment time, we conducted a sub analysis among the 66 treatments performed on patients older than 80 years. Here we registered an acute G2 toxicity in three cases, acute G3 in two cases, late G2 in no cases and late G3 in one case.

## 4. Discussion

MR-Linacs represent a novel technology which has rapidly spread in the recent years, with numerous centers who decided to invest in the field of MRgRT. Despite increasing evidence suggesting superiority of adaptive MRgRT over conventional CT-based RT in some district [14,21], concerns related to a more time- and resources-consuming technology still exist. In order to address these challenging aspects and to optimize MRL utilization, we present a real-world experience of an early-adopter center. With the first treatments performed in 2019, our center was one of the first worldwide to implement the 1.5T MRL for clinical use. Since then, we treated a high volume of patients, with an average of 19 patients treated and 160 fractions delivered per month. With a dedicated team of physician, physicists and radiation therapists, a learning curve could be observed, with 24 patients treated in the first 3 months of the MRL clinical use and 52 in the last 3 months of the period considered for the present analysis. Especially, a clear learning curve could be observed within the first 2 years, with the overall treatment times decreasing of approximately 7 min, being in the past 3 years set on approximately 33 min. The majority of our patients, namely those treated for prostate cancer, could be treated in, on average, 36 min. As expected, longer treatment times were recorded for targets located in the upper abdomen, such as pancreas and liver. In these cases, more time was needed for organs at risk recontouring, e.g., stomach, duodenum and small bowel, for replanning, being that the treatment plans in this district were usually more complex, and for treatment delivery, due to the respiratory motion. Lung tumors, even though affected by longer delivery times due to the respiratory motion, did not require longer times for recontouring and replanning, and could, therefore, be performed, on average, in 37 min. Lymph node metastases, bone metastases and cerebral tumors, generally not affected by significant anatomical inter- or intrafraction variability, displayed the shortest treatment times of our casuistic. In our analyses, 5% of treatment sessions were longer than 1 h. These cases could be considered as outliers, as in our experience they might be due to technical issues where treatments where interrupted and a completion plan was needed.

We present here a large cohort treated for prostate tumors (70% of our analyzed population), within different constellations, namely irradiation of the prostate or prostate bed, including almost 5% of cases with prophylactic pelvic lymph node irradiation. These large field irradiations could be performed without limitations or need to reach targets compromises. Conventionally fractionated treatments represent a small part of our population and were mostly performed in patients with larger fields irradiation. Clinical indications in these cases were more frequently represented by patients with elective nodal pelvic irradiation together with boost targets in the pelvis or by particular anatomical conditions, i.e., few cases of in-field large postoperative seroma. However, in our experience, the advantages of the 1.5T MRL, such as superior soft tissue contrast and adaptive strategies to compensate for interfraction and, more recently with the CMM, intrafraction variability, could be maximally exploited for highly hypofractionated stereotactic treatments of pelvic- and abdominally located targets [22,23].

Very few cases of G3 toxicity, both acute and late, were documented, confirming previously published experiences with optimal safety and excellent tolerability [24,25,26]. Here, we knowledge that our analysis is limited by the retrospective nature of the toxicity investigation. Especially for late toxicity, data were available only for 81% of the patients. This is mainly due to the fact that some patients that referred for prostate treatment to our center from outside the Veneto region did not come back for the regular follow up, eventually limiting the follow up to electronical share of the PSA value, while no valuable information regarding treatment-related toxicity could be collected.

In our cohort patients age ranges from 16 to 90 years. While only 0.5% of the cohort is aged below 40 years, 6% is older than 80 years. Because a 0.5% sample is too small for any consideration, focusing on the 6%, we can observe that the mean treatment time was 37.2 min, therefore very similar to the mean value of the entire cohort, suggesting that the treatment time is influenced majorly by the treatment site, as detailed in Table 3, and not by the age of the patients. In patients aged 80 years or more, even though two cases of acute G3 is almost double the incidence of G3 in the entire cohort (3% vs. 1.6%) and this might indicate increased toxicity in a population of older (and frailer) patients, we believe that the number of events is too small to draw any plausible conclusion.

As we previously reported [4], we generally used 5 mm margins for the PTV, except for the posterior margin for prostate targets, where 3 mm were used. In September 2023, the CMM was implemented in our clinical practice, allowing for target online gating and eventually for treatment plan correction based on an intrafractional drift. This tool represents Elekta Unity’s tracking and automatic gating system and, therefore, a further step toward maximal accuracy of MRgRT at the 1.5T MRL [20].

Nevertheless, some aspects need to be considered. When adopting the CMM, the time for treatment delivery might increment, especially in the following situations: (a) targets located in the upper abdomen where the respiratory-related motion is more pronounced; (b) in case of intrafractional drift, e.g., muscular relaxation for prostate cancer patients, with the consequent need for a baseline-shift plan; and (c) in cases where the margins used for the PTV are tighter, e.g., in our experience, retreatments or other more critical clinical situations [27]. Currently, we need, on average, 35–40 min per fraction. However, our experience suggests that 45–50 min are required to treat targets subjected to a greater intrafractional motion, i.e., upper abdomen. Moreover, because we are going towards a margin reduction strategy, longer overall treatment time might be needed in the future. For instance, a zero-PTV margin strategy is used in the Destination 2 study, an ongoing trial for patients affected by prostate cancer which will be treated with only two fractions at the MR-linac Unity (REC reference 24/EE/0163) with a deescalated dose to the prostate gland and a higher dose to the tumor areas. Therefore, further investigations are warranted, and we are planning next to test the optimal relation between margin reduction and duty cycle. As further step, we plan to develop a protocol to investigate dosimetric and clinical outcomes of treatments performed with reduced margins, in terms of target coverage, local control and toxicity.

## 5. Conclusions

Our real-world large experience as an early-adopter center confirms that 1.5T MRL treatments are feasible for different tumor entities in several anatomical sites. We showed that most of the patients could be treated within 40 min and that the MRL field size did not limited RT indications when larger targets were prescribed. Treatments are safe and were very well tolerated. Protocols for dose escalation and margin reduction, by adopting new comprehensive motion monitoring strategies, are under development.

## Figures and Tables

**Figure 1 cancers-17-02012-f001:**
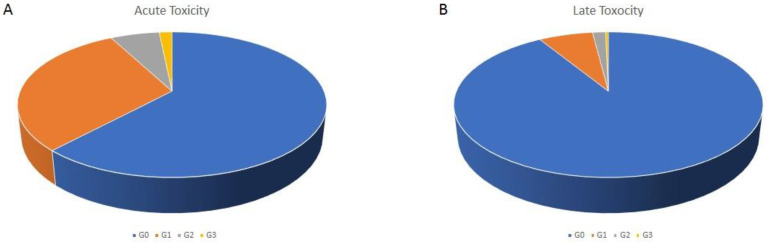
Acute (**A**) and late (**B**) toxicity distribution according to the CTCAE scale.

**Table 1 cancers-17-02012-t001:** Treatment details (n = 1061).

Characteristics	Total	Percentage
RT Indication		
primary tumour/tumour bed	770	72.6%
lymphnodes	170	16.0%
distant metastases	74	7.0%
retreatment	47	4.4%
RT site		
prostate	606	57.1%
prostate bed	108	10.2%
prostate/prostate bed + pelvic lymphnodes	50	4.7%
lypmhnodes	170	16.0%
bone metastases	40	3.8%
pancreas	20	1.9%
liver	12	1.1%
cerebral tumours	11	1.0%
adrenal gland	9	0.8%
lung	8	0.8%
others	27	2.5%
RT dose/fraction schedules		
36.25 Gy in 5 frations	329	31.0%
35 Gy in 5 fractions	300	28.3%
30 Gy in 5 fractions	100	9.4%
60 Gy in 20 fractions	80	7.5%
67.5 Gy in 30 fractions	59	5.6%
36 Gy in 6 fractions	28	2.6%
40 Gy in 5 fractions	24	2.3%
66 Gy in 30 fractions	18	1.7%
25 Gy in 5 fractions	17	1.6%
20 Gy in 5 fractions	10	0.9%
others	96	9.0%
RT delivery schedule			
consecutive days	742	70%	
every other day	308	29%	
single fraction	8	0.08%	
1 fraction/week	3	0.003%	
RT fractions			
total	9076		
ATS	7330	80.8%	
ATP	1746	19.2%	

**Table 2 cancers-17-02012-t002:** Number of treatment courses and average in-room time per fraction per year (n = 1007).

Treatment Time Period	Treatment Courses	Average in-Room Time
10/2019–09/2020	150	40 min
10/2020–09/2021	233	39 min
10/2021–09/2022	215	33 min
10/2022–09/2023	233	34 min
10/2023–06/2024	176	33 min

**Table 3 cancers-17-02012-t003:** Average treatment time according to RT site (n = 1007).

RT Site	Average in-Room Time
pancreas	41 min
liver	39 min
arenal gland	37 min
lung	37 min
prostate/prostate bed +/− pelvic lymphnodes	36 min
lymphnodes	33 min
bone metastases	32 min
cerebral tumours	28 min
others	35 min

**Table 4 cancers-17-02012-t004:** Details of acute toxicity recorded for the patients reporting symptoms (n = 1061).

Grade	Urinary Obstructive Symptoms	Urinary Incontinence	Proctitis	Tenesmus	Sexual Disfunction	Nausea	Others
G1	198	16	54	22	15	10	7
G2	40	0	16	5	2	1	1
G3	16	1	0	0	0	0	0

**Table 5 cancers-17-02012-t005:** Details of late toxicity recorded for the patients reporting symptoms (n = 858).

	Urinary Obstructive Symptoms	Urinary Incontinence	Proctitis	Sexual Disfunction	Others
G1	26	11	9	10	2
G2	3	0	6	5	0
G3	2	0	0	0	1

## Data Availability

The authors confirm that the data supporting the findings of this study are available within the article. Raw data that support the findings of this study are available upon reasonable request from the corresponding author.

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
