# Peer review of "The Clinical Feasibility and Safety of 1.5 T MR-Guided Daily Adapted Radiotherapy in 1000 Patients: A Real-World Large Experience of an Early-Adopter Center"

_cancers, 2025, doi:10.3390/cancers17122012_

Round 1

Reviewer 1 Report

Comments and Suggestions for Authors

General Assessment

The manuscript presents a valuable and well-structured summary of a single-center experience with MR-guided radiotherapy on MR-linacs, covering over 1000 patients. It is clear, relevant, and offers practical insight into clinical feasibility, workflow, and toxicity outcomes. The methodology appears sound, and the conclusions are consistent with the presented data. The study makes a meaningful contribution to the growing body of literature on MR-guided adaptive radiotherapy.

Minor Comments

  1. Please correct minor grammatical issues (e.g., “Technical informations” → “Technical information”).
  2. Consider clarifying the inclusion/exclusion criteria for the selected 1000 patients. It would be helpful to specify whether this number was chosen arbitrarily or based on a predefined analytic or institutional threshold.
  3. Expand on the stratification of treatment groups: What specific criteria were used to assign patients to particular fractionation regimens?
  4. The manuscript mentions the use of a Comprehensive Motion Monitoring (CMM) strategy. Please provide further detail on this protocol and clarify whether it influenced in-room treatment times or workflow optimization.

Suggestions Regarding Results and Methodology

  1. Consider performing a statistical analysis of treatment times stratified by tumor location or treatment technique, using appropriate tests (e.g., t-test, ANOVA).
  2. If feasible, present toxicity data stratified by anatomical site to assess whether specific treatments were associated with different toxicity profiles.
  3. Please clarify how missing data in the late toxicity analysis were handled, given that only 858 out of 1000 patients had available long-term follow-up.
  4. It would be valuable to explore whether any regression analysis was conducted to identify predictors of treatment duration (e.g., age, site, technique).
  5. The authors mention ongoing developments in dose escalation and motion strategies. Even a brief statement on preliminary planning considerations or early insights would enhance the conclusion.

Author Response

The manuscript presents a valuable and well-structured summary of a single-center experience with MR-guided radiotherapy on MR-linacs, covering over 1000 patients. It is clear, relevant, and offers practical insight into clinical feasibility, workflow, and toxicity outcomes. The methodology appears sound, and the conclusions are consistent with the presented data. The study makes a meaningful contribution to the growing body of literature on MR-guided adaptive radiotherapy.

Minor Comments

  1. Please correct minor grammatical issues (e.g., “Technical informations” → “Technical information”).

Thank you, done.

  1. Consider clarifying the inclusion/exclusion criteria for the selected 1000 patients. It would be helpful to specify whether this number was chosen arbitrarily or based on a predefined analytic or institutional threshold.

We decided arbitrarily to include the first 1000 consecutive patients that were treated at Unity from the moment of its installation and permission for clinical use. This is stated at the beginning of the M&M section. We now added the word “arbitrarily” to make clearer, thank you.

  1. Expand on the stratification of treatment groups: What specific criteria were used to assign patients to particular fractionation regimens?

Thank you, we added this sentence to the M&M section: “the fractionation choice was mainly made according to RT site, oncological situation and target/ organs at risk anatomy, e.g. for postoperative prostate treatments eventually including pelvic irradiation as well, slightly-hypofractionated schedules were adopted, for prostate cancer with low/ intermediate risk and low IPSS and a prostatic volume less than 80cc, a 5-fractions schedule was preferred. For upper abdominal targets, i.e. pancreas and liver, 45-50 Gy in 5 or 6 fractions were applied, unless because of target’s size, number of treated lesions and/or anatomical reasons less hypofractionated schedules were needed”.

  1. The manuscript mentions the use of a Comprehensive Motion Monitoring (CMM) strategy. Please provide further detail on this protocol and clarify whether it influenced in-room treatment times or workflow optimization.

Thank you, we added this to the M&M and to discussion: “The CMM represents Elekta Unity’s tracking and automatic gating system, which enables for further treatment precision and accuracy. CMM MR images are available as T1, T2, balanced, FLAIR and SPAIR, whereas the most used in our department are the new T2-weighted Turo Spin Echo sequences for prostate and pelvic located targets and balanced for upper abdomen targets. In order to identify the target on cine images, two template images are firstly generated, with an algorithm very robust for tissue contrast, and registered to the daily 3D MRI. After revision and template approval, cine real time images are acquired using a fast protocol and registered against the template images, having these the same contrast. The target position is real-time compared to the adapted plan. The following CMM structures are identified: 1) the registration structure, which is used by the registration algorithm, and 2) the gating envelope, which encompasses the target, with a margin expansion. A tolerance level is defined and, whenever the gating envelope structure moves outside the registration structures above the tolerance, the beam is disabled and irradiation stops, restarting automatically when the target is again within the tolerance level position. Usually, we defined the GTV or the CTV (e.g. the prostate gland) as registration structure and the PTV as gating envelope. During dose delivery, according to different scenarios, multiple actions can be undertaken, namely: 1) to lower the tolerance lever, 2) to generate a so-called baseline-shift plan or 3) to disable the CMM. Specifically, lowering the tolerance level allows for an increase in the beam hold time, at the cost of lowering treatment precision, while, when a stable shift in the target position occurs, a baseline-shift plan is calculated on the basis of the new position. This is a quickly generated plan (within 1-2 minutes), which takes into account the target coverage, but not all the constraints for the OARs that were implemented in the original or adapted treatment plan. The remaining monitor units are then delivered with this new plan. Finally, disabling completely the CMM leads to the classic motion monitoring strategy, where live MR images with the target structure are visible, but no gating occurs. Based on our preliminary results of the first patients treated with CMM (a manuscript is under preparation, data not shown here) by analysing the beam on-time, the beam-hold counts, the baseline-shift plans, the duty cycle and the overall treatment time, we observed that the implementation of the CMM could support margin reduction strategies as well as higher treatment precision, with only minimal increase in treatment time without affecting patient’s tolerance. For instance, a zero-PTV margin strategy is used in the Destination 2 study, an ongoing trial for patients affected by prostate cancer which will be treated with only 2 fractions at the MR-linac Unity (REC reference 24/EE/0163) with deescalated dose to the prostate gland and a higher dose to the tumour areas.

Suggestions Regarding Results and Methodology

  1. Consider performing a statistical analysis of treatment times stratified by tumor location or treatment technique, using appropriate tests (e.g., t-test, ANOVA).

We thought about it as well, the fact is that we do not feel that this analysis would be informative, for different reasons, such as the fact that the treatment technique was the same, unless if considering ATP vs ATS, where though it is sure that ATP represents a quicker procedure. Regarding the use of the CMM, please refer to the previous comment’s reply. Regarding different tumour locations, as we report on the Table 3, upper abdomen targets requires more time, this is due though to different possible reasons, such as respiratory motion but also to the fact that they often require more time because the monitor units are much more, due to higher dose/ fraction (e.g. 10 Gy) and more modulated treatment plans, due to the vicinity to the bowel, duodenum, stomach and so on. For these reasons we do not feel that a statistical analysis would be particularly informative here.

  1. If feasible, present toxicity data stratified by anatomical site to assess whether specific treatments were associated with different toxicity profiles.

Our population consisted of different primary tumors, among them the only consistent and large population was the prostate, while other district had few patients, so that an analysis per anatomical site could have been not informative.

  1. Please clarify how missing data in the late toxicity analysis were handled, given that only 858 out of 1000 patients had available long-term follow-up.

There is indeed a limitation regarding late toxicity data collection. Unfortunately, we were not able to increase the number of patients available for late toxicity analysis, even though we tried to reach patients by phone or email. We added a sentence to the discussion underlying this limitation.

  1. It would be valuable to explore whether any regression analysis was conducted to identify predictors of treatment duration (e.g., age, site, technique).

We did not for site and technique (please refer to reply to point 1), but we did for age, for this please refer to point number 1 of reviewer 2.

  1. The authors mention ongoing developments in dose escalation and motion strategies. Even a brief statement on preliminary planning considerations or early insights would enhance the conclusion.

Thank you, indeed analysing the beam-on time, the beam-hold counts, the duty cycle and the need of baseline-shift plans, we could observe that the treatment time remained limited, without affecting patient’s compliance and treatment feasibility. Because an original report is under preparation regarding these results, we added here a brief sentence in the discussion (please refer to point number 4 of minor comments), together with a reference to the Destination 2, an ongoing study for prostate cancer treatment at Unity over only 2 fractions.

Reviewer 2 Report

Comments and Suggestions for Authors

This study offers valuable real-world insights into the feasibility and safety profile of MR-Linac treatments across a range of tumor sites, warranting consideration for recommendation. However, to strengthen its conclusions and enhance its impact, the authors are invited to consider the following important issues:

  1. P.3, L.109: the reported age range (16-90 years) is considerable. I recommend to address the potential influence of age on the primary outcomes. Specifically, a discussion of whether age-related subgroup analyses were conducted, or why they were not, is warranted, along with an interpretation of the potential impact of age as a confounding factor on the study findings.

  2. Given that the data were collected between 2019 and 2024, encompassing the period before, during, and after the COVID-19 pandemic, I question whether the potential influence of the pandemic on the observed results has been adequately addressed. Specifically, I recommend exploring whether the data can be analyzed to assess the pandemic’s impact on the outcomes reported in these studies.

  3. As there is potential for increased treatment time with MR-Linac technology and the reported toxicity outcomes, it is crucial to understand the relationship between these factors. Therefore, I recommend exploring the following questions in more detail:

  4. Is there a statistically significant correlation between overall treatment duration per fraction (or total treatment course duration) and the incidence or severity of acute toxicities (specifically Grade 1-3)? This analysis would help determine if longer in-room times are associated with increased toxicity.

  5. Did the time of day (e.g., morning vs. afternoon) that treatment was administered correlate with either treatment duration or toxicity outcomes? Investigating this factor could reveal diurnal variations in patient tolerance or workflow efficiency that might influence treatment outcomes.

  6. Furthermore, some recent studies found differences in either efficacy or toxicity of radiotherapy depending of time of day (circadian phase), or seasons. The authors are invited to address this issue either in Results or in Discussion section.

  7. Keywords are not listed.

Author Response

This study offers valuable real-world insights into the feasibility and safety profile of MR-Linac treatments across a range of tumor sites, warranting consideration for recommendation. However, to strengthen its conclusions and enhance its impact, the authors are invited to consider the following important issues:

  1.  

P.3, L.109: the reported age range (16-90 years) is considerable. I recommend to address the potential influence of age on the primary outcomes. Specifically, a discussion of whether age-related subgroup analyses were conducted, or why they were not, is warranted, along with an interpretation of the potential impact of age as a confounding factor on the study findings.

The reviewer is correct. Analysing further our data though, it appears that only 0,5% of the cohort is aged below 40 years and 6% is older than 80 years. Because a 0,5% samples is too small for any consideration, focusing on the 6% we can observe that the mean treatment time was 37.2 minutes, therefore very similar to the mean value of the entire cohort, suggesting that the treatment time is influenced majorly by the treatment site (as we report in the manuscript) and not by the age of the patients. Regarding toxicity, the 66 treatments performed on patients older than 80 years caused an acute G2 toxicity in 3 cases, acute G3 in 2 cases, late G2 in no cases and late G3 in 1 case. Even though 2 cases of acute G3 is almost double the incidence of G3 in the entire cohort (3% vs 1.6%) and this might indicate increased toxicity in a population of older (and frailer) patients, we believe that the number of events is too small to draw any plausible conclusion. We thank the reviewer for this interesting hint and added a couple of sentences regarding this topic to the results and discussion sections, respectively.

  1.  

Given that the data were collected between 2019 and 2024, encompassing the period before, during, and after the COVID-19 pandemic, I question whether the potential influence of the pandemic on the observed results has been adequately addressed. Specifically, I recommend exploring whether the data can be analyzed to assess the pandemic’s impact on the outcomes reported in these studies.

As the reviewer correctly noticed, the manuscript refers to patients treated before, during and after the COVD-19 pandemic. Our results mostly focus on MR-linac specific aspects (e.g. treatment time, adaption technique...) and while we report toxicity data as well, we do not focus on survival. Even though the COVID-19 pandemic had a dramatic impact on oncological patients, sometimes delaying diagnosis and treatments and having therefore consequences on survival, we do not consider the pandemic a factor that could affect MR-linac specific aspects or the treatment toxicity we reported. Additionally, we are aware that at the beginning of the pandemic papers soliciting the use of hypofractionated schedules (e.g. 5 fractions for adiuvant breast radiotherapy) were published. Because the majority of the treatments at Unity in our department have always been hypofractionated or highly hypofractionated, also this aspect of the pandemic did not affect our indications.  

  1.  

As there is potential for increased treatment time with MR-Linac technology and the reported toxicity outcomes, it is crucial to understand the relationship between these factors. Therefore, I recommend exploring the following questions in more detail:

  1.  

Is there a statistically significant correlation between overall treatment duration per fraction (or total treatment course duration) and the incidence or severity of acute toxicities (specifically Grade 1-3)? This analysis would help determine if longer in-room times are associated with increased toxicity.

Thank you for this very interesting comment. As we did not have >G3 toxicity, we proceeded to analyse the incidence of G2/G3 toxicity for the patients that were treated in less than 38 minutes (our mean in-room time) vs the patients that were treated in 38 minutes or longer and found no difference (7% vs 9%, p=0,1). Even if it might be assumed that a longer treatment time might negatively affect toxicity, mostly increasing the intrafraction variability, it should be noticed that treatments at Unity were performed with the motion monitoring and, after September 2023, with the comprehensive motion monitoring. We added this in the results and discussion sections, respectively.

  1.  

Did the time of day (e.g., morning vs. afternoon) that treatment was administered correlate with either treatment duration or toxicity outcomes? Investigating this factor could reveal diurnal variations in patient tolerance or workflow efficiency that might influence treatment outcomes.

Please refer to the answer to the next questions; because point 5. and 6. seem consistent to each other, we provide a common answer. If, on the other side, this comment refers to the preparation (e.g. empty rectum and full bladder for prostate treatments, at least 3 hours fasting for upper abdominal treatments), patients received a protocol for preparation, so there were not differences between morning and afternoon.

  1.  

Furthermore, some recent studies found differences in either efficacy or toxicity of radiotherapy depending of time of day (circadian phase), or seasons. The authors are invited to address this issue either in Results or in Discussion section.

We are aware that according to some studies, radiation sensitivity might be affected by the circadian rhythm influencing the different phases of the cell cycles. Nevertheless, clinical data regarding this aspect are not totally consistent, some of them addressing toxicities and outcomes being more favourable when patients were treating in the morning/ afternoon and others when patients were treated in the evening. Secondly, we treated all our patients in the morning or early afternoon, being our working day for clinical treatments at Unity approximately 8am-4pm. Finally, the consideration about the circadian cell phases affecting RT efficacy or toxicity might be applied to all type of radiotherapy and it does not appear to be MR-linac specific, as our manuscript is intended to be. For these reasons, we thank for the comment, but we feel that adding analyses or considerations about this topic would go beyond the purpose of the manuscript.

  1.  

Keywords are not listed.

Thank you, we now added the following keywords: MR-Linac, Adaptive radiotherapy, MR-guided radiotherapy, Online plan adaptation

Reviewer 3 Report

Comments and Suggestions for Authors

The presented manuscript briefly summarizes the authors' experience in magnetic resonance radiotherapy (MRI) on MR linear accelerators at the Oncology Center of the IRCCS Sacro Cuore Don Calabria Hospital. The sample included 1,000 patients with various types of cancer (mainly prostate), who underwent 1,061 courses of treatment between October 2019 and June 2024. The data presented looks very impressive, although of course it would be nice to have more specific information on side effects for different types of cancer. However, given the authors' publication activity regarding preliminary results, there is great hope to see a more detailed analysis soon. 

In my opinion, the manuscript is almost ready for publication, except for a number of technical details that need to be resolved.

  1. Keywords.
  2. Poor quality of tables.
  3. Authors' contributions.
  4. Removing non-manuscript parts remaining from the template.
  5. Refine Ref. 16.

Author Response

The presented manuscript briefly summarizes the authors' experience in magnetic resonance radiotherapy (MRI) on MR linear accelerators at the Oncology Center of the IRCCS Sacro Cuore Don Calabria Hospital. The sample included 1,000 patients with various types of cancer (mainly prostate), who underwent 1,061 courses of treatment between October 2019 and June 2024. The data presented looks very impressive, although of course it would be nice to have more specific information on side effects for different types of cancer. However, given the authors' publication activity regarding preliminary results, there is great hope to see a more detailed analysis soon. 

In my opinion, the manuscript is almost ready for publication, except for a number of technical details that need to be resolved.

  1. Keywords.

We now added the following keywords: MR-Linac, Adaptive radiotherapy, MR-guided radiotherapy, Online plan adaptation

  1. Poor quality of tables.

Table quality was improved.

  1. Authors' contributions.

Authors‘ contributrion have been included in the revised manuscript.

  1. Removing non-manuscript parts remaining from the template.

Non-manuscript part werew removed. Thank you

  1. Refine Ref. 16.

Ref 16 was edited. Thank you